# Patterns of contraceptive use through later reproductive years: A cohort study of Australian women with chronic disease

**Melissa L. Harris**[1,2]*, **Nicholas Egan**[1,2], **Peta M. Forder**[1,2], **Deborah Bateson**[3,4], **Deborah Loxton**[1,2]

**1** Centre for Women's Health Research, College of Health, Medicine and Wellbeing, The University of Newcastle, Newcastle, New South Wales, Australia, **2** Hunter Medical Research Institute, Newcastle, New South Wales, Australia, **3** Family Planning NSW, Ashfield, New South Wales, Australia, **4** Discipline of Obstetrics, Gynaecology and Neonatology, Faculty of Medicine and Health, University of Sydney, Sydney, Australia

* Melissa.Harris@newcastle.edu.au

**Data Availability Statement:** The data used as part of this analysis are bound by ethical restrictions due to containing person level data. These restrictions have been imposed by the respective

## Abstract

### Background

Pregnancies among women with chronic disease are associated with poor maternal and fetal outcomes. There is a need to understand how women use or don't use contraception across their reproductive years to better inform the development of preconception care strategies to reduce high risk unintended pregnancies, including among women of older reproductive age. However, there is a lack of high-quality longitudinal evidence to inform such strategies. We examined patterns of contraceptive use among a population-based cohort of reproductive aged women and investigated how chronic disease influenced contraceptive use over time.

### Methods and findings

Contraceptive patterns from 8,030 women of reproductive age from the Australian Longitudinal Study on Women's Health (1973–78 cohort), who were at potential risk of an unintended pregnancy were identified using latent transition analysis. Multinomial mixed-effect logistic regression models were used to evaluate the relationship between contraceptive combinations and chronic disease. Contraception non-use increased between 2006 and 2018 but was similar between women with and without chronic disease (13.6% vs. 12.7% among women aged 40–45 years in 2018). When specific contraceptive use patterns were examined over time, differences were found for women with autoinflammatory diseases only. These women had increased odds of using condom and natural methods (OR = 1.20, 95% CI = 1.00, 1.44), and sterilisation and other methods (OR = 1.61, 95% CI = 1.08, 2.39) or no contraception (OR = 1.32, 95% CI = 1.04, 1.66), compared to women without chronic disease using short-acting methods and condoms.

### Conclusion

Potential gaps in the provision of appropriate contraceptive access and care exist for women with chronic disease, particularly for women diagnosed with autoinflammatory

data custodians for the administrative datasets including the Department of Health, Medicare Australia and the Australian Institute of Health and Welfare. For further information regarding access to Australian Longitudinal Study on Women's Health data, including linked datasets, requests are to be sent to info@alswh.org.au.

**Funding:** The research on which this paper is based was conducted as part of the Australian Longitudinal Study on Women's Health (ALSWH) by the University of Newcastle and The University of Queensland and funded by the Australian Government Department of Health and Aged Care (G1700929). Dr Melissa L. Harris is funded through an Australian Research Council Discovery Early Career Researcher Award (DE190101134). The funders had no role in study design, data collection and analysis, decision to publish, or preparation of the manuscript.

**Competing interests:** The authors have declared that no competing interests exist.

conditions. Development of national guidelines as well as a clear coordinated contraceptive strategy that begins in adolescence and is regularly reviewed during care management through their main reproductive years and into perimenopause is required to increase support for, and agency among, women with chronic disease.

## Introduction

Unintended pregnancy is highest among young women, but there is also a growing consensus that it is a significant public health issue for women of older reproductive age [1, 2]. In addition, the prevalence of chronic disease among women of childbearing age is on the rise. In Australia it is estimated that around 30% of women will be diagnosed with at least one chronic health condition during their reproductive years [3]. This trend is predicted to increase over the coming decade, with chronic disease prevalence increasing substantially across successive generations. For women with chronic disease, unintended pregnancies are associated with serious adverse maternal and perinatal outcomes such as congenital abnormalities, early pregnancy loss, and stillbirth [4–7]. Use of effective contraception is recommended as a key strategy to allow these women to plan pregnancies for times of optimal health, or to provide sufficient time to discontinue potentially teratogenic medications and switch to medications which have greater compatibility with pregnancy. However, contraceptive counselling among chronic disease populations remains low [8, 9]. Given chronic disease is on the rise and these women report unintended pregnancy rates at similar or higher rates than the general population, understanding how they use or don't use contraception is critical to tailoring contraceptive counselling interventions for women with chronic disease as they move through their childbearing years [8, 10, 11].

Despite this, there is limited population-level evidence regarding the contraceptive practices of women with chronic disease, and no studies have examined contraceptive patterns using nationally representative longitudinal data. Of the few available studies, the findings have been equivocal, driven by a reliance on retrospective cross-sectional study designs, differences in contraceptive methods examined and a focus on small single disease samples [10, 12, 13]. As such, the prevalence of contraceptive use and types of methods employed have varied widely across chronic disease populations with contraceptive use found to range from around 30% to as high as 99% [14–16]. Only one study has longitudinally examined contraceptive use among women with chronic disease of childbearing age using state-based insurance claims data [17]. While they found only one-third of women with chronic disease were prescription contraceptive users (compared to 40% of women without a chronic condition), they were unable to examine a range of contraceptive options and they failed to account for women not at risk of pregnancy. Further, while previous studies have involved women with chronic disease across the reproductive life span, recent Australian research suggests that contraceptive patterns differ markedly by life stage and over time [18, 19]. It is therefore important to take a lifecourse approach to contraceptive use among women with chronic disease to understand contraceptive practices as women transition through their childbearing years. This study aimed to establish an evidence-base regarding the contraceptive practices of women with chronic disease by examining patterns of contraceptive use over time among an Australian cohort of women born 1973–78 who have been prospectively followed for over 20 years.

## Materials and methods

### Overview of study design

Data were obtained from the 1973–78 cohort of the Australian Longitudinal Study on Women's Health (ALSWH), a national population-based study examining health and wellbeing among Australian women. Women were randomly sampled through the national health insurer's database (Medicare). This cohort has been found to be largely representative of the population of women in this age group [20]. These women have completed surveys in 1996, 2000 and then on a three-yearly schedule thereafter.

### Participants

This retrospective analysis focused on women who completed Surveys 4, 6 or 8 conducted in 2006 (aged 28–33 years), 2012 (aged 34–39 years) and 2018 (aged 40–45 years). These time points provided measurements across women's main reproductive years. Of the 14,247 women who completed the baseline survey in 1996, 9,604 women were eligible for linked data analysis and completed the questions related to contraceptive use at the selected analysis time points (Fig 1). At each survey, women were considered not at risk of an unintended pregnancy if they reported any of the following: no male partner, hysterectomy, currently pregnant, trying to become pregnant, infertile partner, or partner with low or zero sperm count. Excluding women who were not at risk at all time points (N = 1,574), the final sample for analysis included 8,030 women. Women included in this analysis had a similar demographic profile to the full 1973–78 cohort at baseline in 1996 (S1 Table).

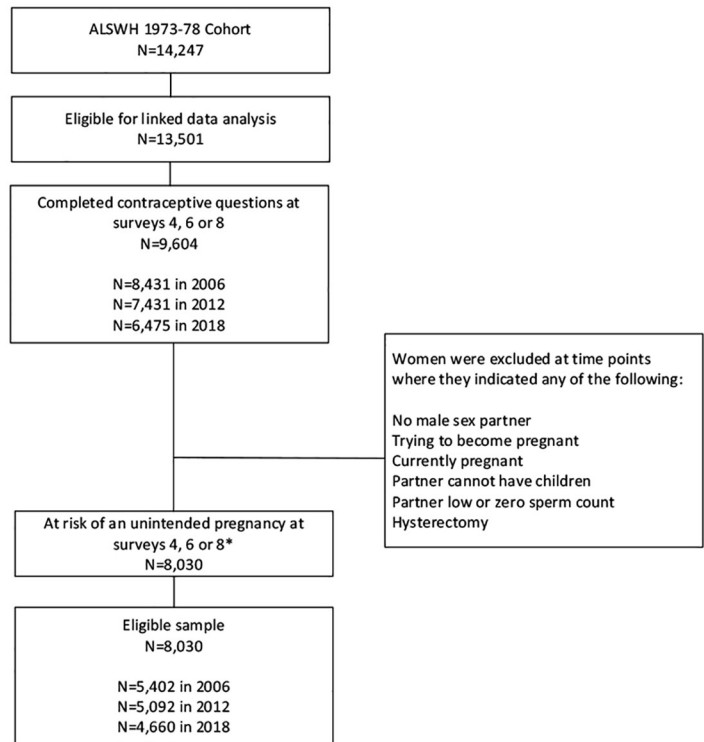

**Fig 1. Determination of eligible sample.** *Percentage of women at each survey who completed contraceptive questions but were excluded due to being not at risk of an unintended pregnancy: 2006 = 36%, 2012 = 31%, 2018 = 28%.

## Measures

**Contraceptive use.** Contraceptive use was measured at each survey. Contraceptive patterns were derived from the question "What forms of contraception do you use now?" At each survey, contraception was measured with 14 response options (participants were able to select more than one option). Women were also asked if they had had a tubal sterilisation or if their partner had had a vasectomy. For this analysis, responses were collapsed into eight groups based on contraceptive efficacy: sterilisation (tubal sterilisation or vasectomy); long-acting reversible contraception (LARC; the progestogen intrauterine system, the copper intrauterine device and the progestogen-only implant); short-acting hormonal contraception (progestogen-only contraceptive pill, combined oral contraceptive pill, oral contraception of unknown type, vaginal ring and depot injection); condoms; natural methods (withdrawal and fertility-based awareness methods); emergency oral contraception; other contraception; and no contraception [19].

**Chronic diseases.** Chronic disease was measured at each survey by the presence or absence of seven physical chronic diseases that have been associated with poor maternal and fetal outcomes. These included diabetes, cardiac disease (including hypertension), asthma, autoinflammatory arthropathies and connective tissue disease (e.g., rheumatoid arthritis [RA] and systemic lupus erythematosus [SLE]), inflammatory bowel disease (IBD), multiple sclerosis, and thyroid disease). Chronic diseases were ascertained using multiple data sources (self-report survey, state-based hospital admissions data, pharmaceutical prescriptions and disease-specific medical claims from general practitioners and specialist care) and employing disease-specific algorithms (developed in concert with clinical experts) to increase chronic disease accuracy in this population. Detailed information on the methods employed are described elsewhere [21].

**Covariates.** Time-varying covariates were measured at each survey. Sociodemographic variables included age, highest educational qualification (no formal qualifications, school/higher school certificate; trade/certificate/diploma; university/higher degree), area of residence (categorised according to the Accessibility/Remoteness Index of Australia (ARIA+) classification system as: major cities; inner regional; outer regional/remote/very remote), relationship status (partnered; unpartnered), and country of birth (Australia; other English speaking; other).

Health care card holder status (a concession card provided for government-subsidised health care) was also included as a surrogate for socioeconomic status (yes; no). Health-related factors included smoking status (current smoker; ex-smoker; non-smoker) and body mass index (underweight [$<18.5$ kg/m$^2$]; healthy [$\geq18.5$ and $<25$ kg/m$^2$]; overweight [$\geq25$ and $<30$ kg/m$^2$]; obese [$\geq30$ kg/m$^2$]) [22].

Reproductive health factors included history of pregnancy (yes; no) and history of pregnancy termination (yes; no). Given that contraception is often used for non-contraceptive reasons, the presence or absence of self-reported gynaecological conditions such as polycystic ovarian syndrome and endometriosis (yes; no) as well as the frequent experience of menstrual symptoms such as irregular periods, heavy period or severe period pain were also included (yes; no).

## Ethics

All data for this project were obtained from the ALSWH (see www.alswh.org.au/ for further details), approved under Expression of Interest process (EOI A696) and provided in de-identified form. This project has ongoing ethical clearance from the University of Newcastle and University of Queensland's Human Research Ethics Committees. Ethical approval for linkage

of ALSWH survey data to the Admitted Patients Data Collections (APDC) was received from the NSW Population and Health Services Research Ethics Committee and other equivalent state and territory-based committees. Linkage to the National Death Index (NDI) was approved by the Australian Institute of Health and Welfare Ethics Committee. Women provided explicit written consent to participate in the ALSWH as well as for linkage to de-identified administrative health records prior to 2005. From 2005, an 'opt-out' consent process was approved by the data custodians and relevant ethics committees for data linkage, with participants regularly reminded of this process. ALSWH participants who decline health record linkage are excluded from data linkage requests.

## Statistical analysis

Contraceptive data were entered into separate latent transition models evaluating three to eight latent statuses. Establishing the optimal LTA model was conducted in a similar fashion to Harris et al. [19] and was based on clinical interpretability, latent class separation and goodness-of-fit statistics. Women were assigned a latent status at each survey they completed where they were at risk of an unintended pregnancy, up to a maximum of three time points. These latent statuses could be the same or vary over time. The probabilities of women transitioning being latent statuses over time are presented in S2 Table. Each latent status described one or more contraceptives being used concurrently. For example, Status 1 ("short-acting and condom") refers to women who used short-acting contraceptive methods and condoms simultaneously. A classify-analyse approach was then used to assign each participant to a latent status at each time point, according to the latent status with the greatest posterior probability. Latent transition analysis was performed using PROC LTA procedure (The Methodology Centre, Penn State) in SAS 9.4 software (SAS Institute Inc). The number of time points with latent statuses with low probabilities (<2%) were minimised as this would contribute to numerical estimation issues in the subsequent regression models. Multinomial mixed-effect logistic regression models were developed, with the assigned latent status as the multinomial outcome, participant ID as a random effect to account for correlation between observations on the same participant over time, while controlling for time-varying covariates as fixed effects. We performed a complete case analysis with participants with missing data omitted from the analysis. Models were developed for the presence of any chronic condition and separately for each chronic disease.

## Results

### Sample characteristics

In 2006, 25.8% of women (aged 28–33 years) reported at least one chronic disease. This increased to 35.5% by 2018 when aged 40–45 years (Table 1). The most common chronic condition was asthma (17.9% in 2006). Other conditions, including diabetes, cardiac disease, autoinflammatory arthropathies and connective tissue disease, IBD, multiple sclerosis, and thyroid disease each had prevalence rates of less than 5% in 2006. However, the prevalence of each of these conditions had approximately doubled by 2018 when the women were aged 40–45 years. Given the low frequencies of autoinflammatory conditions (AICs; i.e., autoinflammatory arthropathies and connective tissue disease, IBD, multiple sclerosis, and thyroid disease) these were combined into a single autoinflammatory disease category for subsequent modelling.

Comparing women with and without chronic disease in this sample, there were few differences across sociodemographic, health behaviour and reproductive health characteristics at Survey 4 in 2006 (Table 2). Women with chronic disease were less likely to have a university degree (42.5% vs. 48.0%), and more likely to report their ability to manage on available income

**Table 1. Proportion of women with chronic disease over the observation period.**

| Chronic disease | 2006 Aged 28–33 N = 5,402 | 2012 Aged 34–39 N = 5,092 | 2018 Aged 40–45 N = 4,660 |
|---|---|---|---|
| | n (%) | n (%) | n (%) |
| Any physical chronic disease | 1,394 (25.8) | 1,636 (32.1) | 1,656 (35.5) |
| Diabetes | 144 (2.7) | 210 (4.1) | 246 (5.3) |
| Cardiac disease | 260 (4.8) | 396 (7.8) | 430 (9.2) |
| Asthma | 965 (17.9) | 928 (18.2) | 883 (18.9) |
| [a]Autoinflammatory disease | 276 (5.1) | 469 (9.2) | 510 (10.9) |
| Arthropathies | 97 (1.8) | 150 (2.9) | 186 (4.0) |
| Inflammatory bowel disease | 28 (0.5) | 40 (0.8) | 53 (1.1) |
| Thyroid disease | 45 (0.8) | 155 (3.0) | 194 (4.2) |
| Multiple sclerosis | 85 (1.6) | 120 (2.4) | 116 (2.5) |

[a]Includes autoinflammatory arthropathies, inflammatory bowel disease, thyroid disease and multiple sclerosis; no missing data for chronic disease variables as these were derived from a range of data sources.

as impossible or difficult always (16.8% vs. 11.7%). Similar differences were also observed in 2018 when the women were aged 40–45 years.

## Trends in contraceptive use

In 2006, 91.5% of women with chronic disease were using some form of contraception compared to 93.2% of women without chronic disease (Table 3). At this time point, 43.8% of women with chronic disease were using short-acting hormonal contraception compared to 46.2% of women without chronic disease. By 2018, these proportions had reduced, but were similar (15.1% vs 15.6%). In 2006, 6.5% and 5.3% of women with and without chronic disease were using LARC, respectively. By 2018, LARC use had increased substantially with 23.7% of women with chronic disease using this method compared to 20.7% of women without chronic disease. A similar increase was noted for sterilisation methods, with relatively low use in 2006 (11.3% vs 9.0%, respectively) and substantially higher use at age 40–45 years in 2018 (31.6% vs 33.0%, respectively).

Among women with chronic disease, use of one contraceptive method only was similar over time (70.0% and 71.9% in 2006 and 2018 respectively). Among women with chronic disease, use of two or more contraceptive methods declined from 21.5% in 2006 to 14.5% in 2018. Contraceptive non-use increased over the observation period with 13.6% of women with chronic disease and 12.7% without chronic disease not using any contraception in 2018 (when aged 40–45 years). Emergency contraception was reported with very low frequencies across time for women with and without chronic disease (1.6% vs 1.1% in 2006 and 0.5% vs 0.2% in 2018, respectively). Given these low frequencies and the purpose of emergency contraception being to prevent pregnancy where contraception has not been used, misused, or has failed, it was excluded from further analysis.

## Contraceptive combinations

A five-status LTA model was selected as the optimal model, given its reasonable clinical interpretability and goodness-of-fit (S3 Table). Status 1 (Table 4), "short-acting and condom" was characterised by high use of short-acting methods (100% probability) with some supplementation with condoms (15% probability). Status 2, "condom and natural" captured high use of condoms (69% probability) with some supplementation of non-hormonal natural methods (38% probability) and other methods (8% probability). Status 3, "sterilisation and other" was

**Table 2. Characteristics of women from the 1973–78 cohort at Survey 4 in 2006 (aged 28–33 years), according to chronic disease status (n = 5,402).**

| Characteristic | Category | Chronic disease status | |
|---|---|---|---|
| | | **Absent n = 4,008 n (%)** | **Present n = 1,394 n (%)** |
| **Sociodemographics** | | | |
| Country of birth | Australia | 3,548 (88.9) | 1,264 (90.9) |
| | Other English-speaking background | 181 (4.5) | 69 (5.0) |
| | Non-English-speaking background | 261 (6.5) | 58 (4.2) |
| Area of residence | Major cities | 2,301 (57.4) | 763 (54.7) |
| | Inner regional | 985 (24.6) | 401 (28.8) |
| | Outer regional/remote/very remote | 722 (18.0) | 230 (16.5) |
| Education | No formal qualifications | 30 (0.8) | 23 (1.7) |
| | School certificate/higher school certificate | 986 (24.7) | 361 (26.0) |
| | Trade/apprentice/certificate/diploma | 1,056 (26.4) | 412 (29.7) |
| | University/higher degree | 1,925 (48.2) | 593 (42.7) |
| Relationship status | Partnered | 3,228 (80.8) | 1,092 (78.7) |
| | Unpartnered | 768 (19.2) | 296 (21.3) |
| Health care card | No | 3,520 (87.9) | 1,137 (81.6) |
| | Yes | 486 (12.1) | 256 (18.4) |
| **Health factors** | | | |
| Smoking | Non-smoker | 2,336 (58.6) | 760 (54.7) |
| | Ex-smoker | 870 (21.8) | 303 (21.8) |
| | Current smoker | 783 (19.6) | 326 (23.5) |
| Body mass index | Underweight | 137 (3.5) | 31 (2.3) |
| | Healthy | 2,329 (59.5) | 646 (47.7) |
| | Overweight | 910 (23.3) | 359 (26.5) |
| | Obese | 535 (13.7) | 318 (23.5) |
| **Reproductive health** | | | |
| History of pregnancy | No | 3,458 (86.8%) | 1,186 (85.3%) |
| | Yes | 526 (13.2%) | 204 (14.7%) |
| History of termination | No | 3,320 (82.9%) | 1,118 (80.3%) |
| | Yes | 687 (17.1%) | 275 (19.7%) |
| Parity | Zero | 1,833 (45.7%) | 621 (44.5%) |
| | One | 797 (19.9%) | 292 (20.9%) |
| | Two | 975 (24.3%) | 345 (24.7%) |
| | Three or more | 403 (10.1%) | 136 (9.8%) |
| Menstrual symptoms | No | 3,397 (85.4%) | 1,105 (79.6%) |
| | Yes | 580 (14.6%) | 283 (20.4%) |
| History of PCOS* | No | 3,942 (98.4%) | 1,352 (97.0%) |
| | Yes | 66 (1.6%) | 42 (3.0%) |
| History of endometriosis | No | 3,893 (97.1%) | 1,342 (96.3%) |
| | Yes | 115 (2.9%) | 52 (3.7%) |

*PCOS = Polycystic ovary syndrome; missing data for each variable was less than 0.5%, except for history of pregnancy (0.52%), menstrual symptoms (0.68%) and body mass index (2.52%).

dominated by vasectomy or tubal sterilisation (100% probability) but included supplementation of other methods for some women (16% probability). Status 4, "LARC" included the use of long-acting methods (100% probability), with a small amount of supplementation such as condoms (3% probability). The "no contraception" status (Status 5) captured the absence of

**Table 3. Contraceptive trend over time by chronic disease status.**

| Contraception | | Chronic disease status 2006 (Survey 4) Aged 28–33 years | | Chronic disease status 2012 (Survey 6) Aged 34–39 years | | Chronic disease status 2018 (Survey 8) Aged 40–45 years | |
|---|---|---|---|---|---|---|---|
| | | No n = 4,008 n (%) | Yes n = 1,394 n (%) | No n = 3,456 n (%) | Yes n = 1,636 n (%) | No n = 3,004 n (%) | Yes n = 1,656 n (%) |
| Any contraception | | 3,737 (93.2) | 1,276 (91.5) | 3116 (90.2%) | 1463 (89.4%) | 2,622 (87.3) | 1,431 (86.4) |
| Condom | | 1,297 (32.4) | 423 (30.3) | 899 (26.0%) | 419 (25.6%) | 534 (17.8) | 261 (15.8) |
| Short acting[A] | | 1,853 (46.2) | 611 (43.8) | 1001 (29.0%) | 443 (27.1%) | 455 (15.1) | 259 (15.6) |
| LARC[B] | | 213 (5.3) | 91 (6.5%) | 486 (14.1%) | 248 (15.2%) | 622 (20.7) | 393 (23.7) |
| Natural methods[C] | | 503 (12.5) | 172 (12.3) | 473 (13.7%) | 186 (11.4%) | 337 (11.2) | 177 (10.7) |
| Sterilisation[D] | | 361 (9.0) | 157 (11.3) | 749 (21.7%) | 391 (23.9%) | 991 (33.0) | 523 (31.6) |
| Other methods | | 347 (8.7) | 145 (10.4) | 104 (3.0%) | 57 (3.5%) | 116 (3.9) | 73 (4.4) |
| Emergency | | 43 (1.1) | 22 (1.6) | 21 (0.6%) | 14 (0.9%) | 6 (0.2) | 9 (0.5) |
| No contraception | | 271 (6.8) | 118 (8.5) | 340 (9.8%) | 173 (10.6%) | 382 (12.7) | 225 (13.6) |
| Number of contraceptives | 0 | 271 (6.8) | 118 (8.5) | 340 (9.8%) | 173 (10.6%) | 382 (12.7) | 225 (13.6) |
| | 1 | 2,962 (73.9) | 976 (70.0) | 2558 (74.0%) | 1207 (73.8%) | 2,204 (73.4) | 1,191 (71.9%) |
| | 2 | 713 (17.8) | 280 (20.1) | 522 (15.1%) | 236 (14.4%) | 404 (13.4) | 228 (13.8%) |
| | 3+ | 62 (1.5) | 20 (1.4) | 36 (1.1%) | 20 (1.2%) | 14 (0.5) | 12 (0.7%) |

[A] The short-acting category was composed of the pill (91.5%), the minipill (5.8%), injection (2.5%) and vaginal ring (0.7%).

[B] The long-acting reversible contraception (LARC) category was composed of progestogen-only IUD (64.5%), implant (32.2%) and copper IUD (4.0%).

[C] The natural methods category was composed of withdrawal method (88.9%) and fertility awareness methods (20.1%).

[D] The sterilisation category was composed of vasectomy (79.1%) and tubal ligation (21.6%).

Note: Types of contraception do not add to 100% as respondents were able to select multiple methods.

LARC = long-acting reversible contraception.

contraceptive use (100% probability). Status 1, "short-acting and condom", was selected as the reference status as it was comprised of two of the most popular contraceptive methods among reproductive-aged women, the pill and condom. This was reflected in the data, with 44% of women in 2006 belonging to Status 1 in 2006 (S4 Table). As such, Status 1 was appropriate as a baseline contraceptive against which to make comparisons in the subsequent modelling.

## Contraceptive use patterns over time

Women were most likely to remain in the same latent status between time points (S2 Table), with women in Status 3 ("sterilisation and other") in 2006 the most likely to continue using the

**Table 4. Probability of individual contraception contributing to contraceptive patterns over time for Australian women born 1973–78, using a five-status LTA model.**

| Latent Status | Latent status description | Item-response probabilities for each status | | | | | | |
|---|---|---|---|---|---|---|---|---|
| | | Condom | Short-acting | LARC | Natural | Other | Sterilisation | No contraception |
| Status 1 | Short-acting and condom | 0.15 | 1.00 | - | 0.03 | - | - | - |
| Status 2 | Condom and natural | 0.69 | 0.04 | - | 0.38 | 0.08 | - | - |
| Status 3 | Sterilisation and other | 0.02 | 0.04 | 0.06 | - | 0.16 | 1.00 | - |
| Status 4 | LARC | 0.03 | - | 1.00 | - | - | 0.01 | - |
| Status 5 | No contraception | - | - | - | - | - | - | 1.00 |

Note: Dashed cells have item-response probabilities <0.01. Shaded status (#1, short-acting and condom) was selected as the reference class for subsequent analysis.

Natural methods = withdrawal and fertility-awareness methods.

LARC = Long-acting reversible contraception.

same contraception in 2012 (probability = 0.86). Women in Status 5 in 2006 ("no contraception") were most likely to remain in Status 5 when measured again in 2012 (probability = 0.32) but had a moderate probability of transitioning to Status 2 ("condom and natural", probability = 0.26) or Status 3 ("sterilisation and other", probability = 0.18). Women in Status 1 ("short-acting and condom) in 2012 had similar probabilities of transitioning to "condom and natural" (probability = 0.14), "sterilisation and other" (probability = 0.17), "LARC" (probability = 0.15)" and "no contraception" (probability = 0.13).

### Contraceptive use by women with chronic disease

There was no evidence to support a difference in the patterns of contraceptive use for women with any chronic disease compared to women without chronic disease (Table 5). When each of the chronic conditions were examined separately, a difference in contraceptive patterns was observed only for women with autoinflammatory disease. Women with autoinflammatory disease had increased odds of using condom and natural methods (OR = 1.20, 95% CI = 1.00, 1.44), and sterilisation and other methods (OR = 1.61, 95% CI = 1.08, 2.39) or no contraception (OR = 1.32, 95% CI = 1.04, 1.66), compared to women without chronic disease using short-acting methods and condoms (full model results are presented in S5–S9 Tables).

## Discussion

### Main findings

By examining contraceptive combinations for women with chronic disease who were at risk of an unintended pregnancy and following these women over a 12-year observation period, we were able to provide an accurate account of contraceptive use (and non-use) for women with diabetes, cardiac disease, AICs and asthma. Although women with chronic disease used contraception at similar rates to women without chronic disease in the community, around 30% were either non-users of contraception or users of low efficacy contraception by age 40–45. However, when individual chronic diseases were examined, there was evidence to suggest that women with AICs were more likely to engage in low efficacy contraception or no contraception compared to other women. Such practices place these women at increased risk of high-risk unintended pregnancy. As such, these findings have the potential to influence the

**Table 5. Multinomial mixed-effect models for the effect of chronic disease status on contraceptive use for Australian women, aged 28 to 45 across three time points (2006, 2012 & 2018).**

| Model | Chronic disease status | Condom & natural OR (95% CI) | Sterilisation & other OR (95% CI) | [a]LARC OR (95% CI) | No contraception OR (95% CI) |
|---|---|---|---|---|---|
| 1 | Any physical chronic disease | 0.97 (0.88, 1.08) | 1.23 (0.96, 1.57) | 1.12 (0.96, 1.32) | 1.07 (0.93, 1.23) |
| 2 | Cardiac disease | 0.86 (0.67, 1.10) | 0.79 (0.52, 1.20) | 1.03 (0.78, 1.34) | 0.86 (0.67, 1.10) |
| 3 | Diabetes | 1.28 (0.95, 1.74) | 0.84 (0.48, 1.47) | 1.34 (0.95, 1.91) | 1.28 (0.95, 1.74) |
| 4 | Asthma | 1.00 (0.85, 1.18) | 1.14 (0.84, 1.54) | 1.04 (0.86, 1.26) | 1.00 (0.85, 1.18) |
| 5 | Autoinflammatory disease | 1.32 (1.04, 1.66) | 1.61 (1.08, 2.39) | 1.18 (0.91, 1.53) | 1.32 (1.04, 1.66) |

Reference status = short-acting and condom; reference level for disease = disease not present.

Each model controlled for age, country of birth, area of residence, highest educational qualification, relationship status, health care card holder status, smoking status, body mass index, history of pregnancy, history of termination, history of miscarriages, menstrual symptoms, history of polycystic ovary syndrome, history of endometriosis, and survey wave.

LARC = Long-acting reversible contraception.

Full model results can be found in S5–S9 Tables.

development of targeted clinical interventions and guidelines to help support provision of effective contraception for women with AICs.

## Interpretation

Overall, contraceptive use among women with chronic disease in our study was found to be relatively high, with rates reported above 85% across the observation period. This finding is supported by one population-based study but contrasts with previous cross-sectional Australian and short-term longitudinal international research (although the Australian study found similar rates between those with and without chronic disease) [8, 17, 23]. Encouragingly, there were substantial increases in the uptake of LARC and permanent methods by the time women were 40–45 years. Use of highly effective methods among women with chronic disease is supported by a previous population-based international study, although that finding was largely driven by sterilisation [23]. While the use of permanent methods in our study increased over time, with around a third of women using these methods by age 40–45, almost one-quarter were using LARC by this time. This is important as these methods are safe for most women with chronic disease, have low failure rates and provide additional benefits during perimenopause [24]. LARC use among women with chronic disease in this cohort however was found to be substantially lower than that reported by women with chronic disease in the 1989–95 cohort at similar ages and suggests that there may be a generational shift in the perceptions surrounding the use of LARC, including the suitability of these methods for women of reproductive age with chronic disease [25].

When focused on patterns of contraceptive use across chronic disease groups, only women with AICs were found to differ in their contraceptive practices to women without chronic disease who used short-acting methods and condoms. Interestingly, method choice was complex amongst this group, with these women more likely to engage in permanent contraception or alternatively, low efficacy methods and no contraception. In this cohort, sterilisation was driven by partner vasectomy [19]. For women of older reproductive age with AICs who have completed their families or do not wish to have children, this finding is promising given that partner vasectomy is a more straight forward procedure with fewer risks than female sterilisation [26, 27].

Concerning, however, is the increased use of low efficacy methods and non-contraceptive use among women with AICs. Previous international research has found 70–80% of women of reproductive age with AICs were non-users of prescription contraception [16, 17]. Similarly, a Brazilian study found that while women with SLE were relatively high users of hormonal contraception prior to their diagnosis, more than half were non-users following their diagnosis, and for those still using contraception, it was most likely condoms [9]. This is despite almost half of the women being on teratogenic medications including methotrexate. Low use of high efficacy contraception among women with AICs using fetotoxic medications has been found by others [28, 29]. Although condoms protect against STIs and are effective at preventing pregnancy when used consistently and correctly, given that this method requires user action with every episode of intercourse, typical contraceptive failure rates for condoms have been estimated to be around 20%. The relatively low efficacy of condoms when used as a sole method of contraception limits their suitability for women with chronic disease, but when used together with a highly effective method such as a LARC or a contraceptive pill they provide protection against STIs and can increase contraceptive protection [30]. However, around half of the women in our study were also predicted to combine their condom use with other low efficacy methods such as withdrawal. The layering of low efficacy methods has been demonstrated among young women, including those with AICs [18, 25]. This is particularly problematic as it has been found that 61% of women with SLE reported using these low efficacy methods and more than half reported having had an unintended pregnancy [31].

Our findings therefore point to a lack of evidence-based advice and support from GPs and specialists. This is particularly important as our recent research has demonstrated that the use of low efficacy contraception has also been found in younger women with AICs [25]. It is essential that individualised contraceptive counselling is included for women of reproductive age from the time of diagnosis and as part of their ongoing care. Switching to low efficacy methods or no contraception after an autoinflammatory diagnosis may be attributed to concerns from health professionals in relation to medical eligibility of combined oral contraceptives and certain AICs, concerns by women themselves, or both [32]. Caution is warranted when prescribing estrogen-containing contraceptives to women with AICs such as RA and SLE due to the elevated risk of venous thromboembolism in those with antiphospholipid syndrome, and in those undergoing IBD-related surgery, with the effectiveness of oral methods reduced by malabsorption [33, 34]. LARC are ideal methods for women with AICs wishing to avoid pregnancy as they are highly effective, are not associated with an increased venous thromboembolism risk and are not impacted by malabsorption. However, misperceptions about IUDs persist, particularly regarding their suitability for young and nulliparous women and the risk of pelvic infections [35, 36]. The copper IUD is appropriate for women with AICs wishing to avoid hormonal methods, although its use in Australia is not currently subsidised by the federal government, unlike progestogen-containing IUDs and implant. Given copper IUDs are suitable for women of all reproductive ages including those in their forties and early fifties as well as for most women with chronic disease, subsidising the copper IUD under the PBS in Australia could facilitate uptake among women with chronic disease.

Despite increasing LARC use being a core outcome of the current Women's Health Strategy currently no formal national guidelines regarding the provision of contraception for women with chronic disease across the reproductive lifecourse exists in Australia [37]. Increased access to, and awareness of current therapeutic guidelines by peak medical associations and key bodies (e.g., eTherapeutic Guidelines) as well as development of referral pathways are required alongside increasing medical education to address the demonstrated lack of expertise and confidence regarding the provision of family planning among GPs and specialists in Australia and other countries [38–40]. Given the increasing prevalence of chronic disease among women of reproductive age, an embedded contraceptive strategy as part of chronic disease management could increase women's agency around contraceptive decision-making. Importantly, although women with diabetes and cardiac disease were found to be using effective contraception at rates similar to the general population, they still require regular review around the suitability of estrogen-containing contraception. The needs of women with cardiac disease are especially difficult to navigate due to the variability in potential risks associated with both contraception type and cardiac disease type and severity. As such, a review of reproductive life plans should be part of best practice management for all women with chronic disease.

## Strengths and limitations

Use of nationally representative longitudinal data is a key strength of this study. We were able to examine a comprehensive set of contraceptive methods (including prescription and non-prescription methods) and applied complex statistical modelling to accurately identify contraception use (including contraceptive combinations). We also considered the dynamic nature of contraceptive use and unintended pregnancy risk across the reproductive lifecourse in our analysis [19, 41]. Given 5,046 women were not at risk of an unintended pregnancy for at least one of the time points, this should be standard practice for future longitudinal contraceptive research. A further strength is our approach to chronic disease measurement [21]. No studies have previously employed such comprehensive methods to ascertain chronic disease among

women of reproductive age or applied these to contraceptive research in the context of chronic disease.

However, as we examined contraceptive use at three time points, six years apart, we were not able to identify switching of contraceptive methods in between these periods. Also, we employed a classify-analyse approach to examine changing chronic disease status as well as other key influencing factors, which potentially introduced a degree of measurement error from the latent status classification. There may have also been some bias in the sample due to differential loss to follow up, however it is unclear whether women with chronic disease would be more or less likely to have completed surveys. Similar to other longitudinal cohort studies, there is an over-representation of tertiary educated women in the ALSWH; however, this is the largest sample examining contraceptive patterns over time for women with chronic disease.

## Conclusion

Across all women, the use of highly effective contraception such as LARC increased over time, although rates were still relatively low and contraceptive non-use also increased. This is problematic and highlights the general need for contraceptive care and counselling to reduce the risks of unintended pregnancy for all women, including those at later reproductive life. We also demonstrated that women with chronic disease take up contraception at similar rates to their same aged peers in the community. However, among women with chronic disease, women diagnosed with AICs were more likely to engage in low efficacy methods of contraception or did not use contraception. Our study therefore highlights the potential gaps in the provision of appropriate contraceptive access and care for women with chronic disease, particularly around the suitability of estrogen-containing contraceptives, including those diagnosed with AICs. The need for the development of national guidelines and a clear contraceptive strategy from adolescence through to menopause are required. Guidelines should encourage regular contraceptive review during routine care as well as training and education for medical professionals to increase support for, and agency among, women with chronic disease. This, in turn, will reduce the occurrence of high-risk unintended pregnancies and facilitate optimal outcomes for planned pregnancies.

## Supporting information

**S1 Table. Comparison of analysed sample to entire cohort at baseline survey (1996).**
(DOCX)

**S2 Table. Latent status transition probabilities (tau estimates) from Time 1 (2006) to Time 2 (2012), and from Time 2 (2012) to Time 3 (2018).**
(DOCX)

**S3 Table. Summary of LTA model diagnostics for the 1973–78 ALSWH cohort.**
(DOCX)

**S4 Table. Latent status membership probabilities (delta estimates) for the five-status model.**
(DOCX)

**S5 Table. Full multinomial mixed-effect models for factors associated with contraceptive use among Australian women (2006–2018), examining the impact of any physical chronic disease.**
(DOCX)

**S6 Table. Full multinomial mixed-effect models for factors associated with contraceptive use among Australian women (2006–2018), examining the impact of chronic cardiac disease.**
(DOCX)

**S7 Table. Full multinomial mixed-effect models for factors associated with contraceptive use among Australian women (2006–2018), examining the impact of diabetes.**
(DOCX)

**S8 Table. Full multinomial mixed-effect models for factors associated with contraceptive use among Australian women (2006–2018), examining the impact of asthma.**
(DOCX)

**S9 Table. Full multinomial mixed-effect models for factors associated with contraceptive use among Australian women (2006–2018), examining the impact of autoinflammatory disease.**
(DOCX)

## Acknowledgments

The research on which this paper is based was conducted as part of the Australian Longitudinal Study on Women's Health by the University of Newcastle and the University of Queensland. We are grateful to the women who provided the survey data. The authors acknowledge the Department of Health and Medicare Australia for providing MBS and PBS data, and the Australian Institute of Health and Welfare (AIHW) as the integrating authority and undertaking the data linkage to the National Death Index (NDI). The authors also acknowledge the following 1) Centre for Health Record Linkage (CHeReL), NSW Ministry of Health and ACT Health for the NSW Admitted Patients and ACT Admitted Patient Care Data Collections; 2) Queensland Health, including the Statistical Services Branch, for the QLD Hospital Admitted Patient Data Collection; 3) Department of Health Western Australia, including the Data Linkage Branch, and the WA Hospital Morbidity Data Collection; 4) SA NT Datalink, SA Health and the Northern Territory Department of Health, for the SA Public Hospital Separations and NT Public Hospital Inpatient Activity Data Collections; 5) The Department of Health Tasmania, and the Tasmanian Data Linkage Unit, for the Public Hospital Admitted Patient Episodes Data Collection; and 6) The Department of Health and Human Services Victoria, Centre for Victorian Data Linkage, for the Victorian Admitted Episodes Dataset.

## Author Contributions

**Conceptualization:** Melissa L. Harris.

**Data curation:** Melissa L. Harris, Nicholas Egan.

**Formal analysis:** Nicholas Egan.

**Funding acquisition:** Melissa L. Harris.

**Investigation:** Melissa L. Harris.

**Methodology:** Melissa L. Harris, Nicholas Egan, Peta M. Forder.

**Project administration:** Melissa L. Harris.

**Visualization:** Nicholas Egan, Peta M. Forder.

**Writing – original draft:** Melissa L. Harris.

**Writing – review & editing:** Melissa L. Harris, Nicholas Egan, Peta M. Forder, Deborah Bateson, Deborah Loxton.

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
