## [Decision Letter · Decision Letter 0]

21 Feb 2023

PONE-D-22-13508Patterns of contraceptive use through later reproductive years: a prospective cohort study of Australian women with chronic diseasePLOS ONE

Dear Dr. Harris,

Thank you for submitting your manuscript to PLOS ONE. After careful consideration, we feel that it has merit but does not fully meet PLOS ONE’s publication criteria as it currently stands. Therefore, we invite you to submit a revised version of the manuscript that addresses the points raised during the review process.

We look forward to receiving your revised manuscript.

Kind regards,

Billy Morara Tsima, MD MSc

Academic Editor

PLOS ONE

Journal Requirements:

Reviewers' comments:

Reviewer's Responses to Questions

**Comments to the Author**

1. Is the manuscript technically sound, and do the data support the conclusions?

Reviewer #1: Yes

Reviewer #2: Yes

2. Has the statistical analysis been performed appropriately and rigorously? 

Reviewer #1: Yes

Reviewer #2: Yes

3. Have the authors made all data underlying the findings in their manuscript fully available?

Reviewer #1: No

Reviewer #2: Yes

4. Is the manuscript presented in an intelligible fashion and written in standard English?

Reviewer #1: Yes

Reviewer #2: Yes

5. Review Comments to the Author

Reviewer #1: Overall, this is a well-written manuscript.

While this study is based on the Australian longitudinal study on women’s health, the current study is clearly retrospective. The title should not include “a prospective cohort study

I recommend clearly stating the study design in the methods section as well. From the description in the methodology section, this is a retrospective cohort study.

For table 2, the presentation of the results should be changed such that the proportions are out of participants with available data. Missing data should not be part of the presented data. Instead, the authors should report on the proportion with available data for each variable. As an example for country of origin (without chronic disease), data is available for 3,990 participants of which 3,548 (88.9%) were born in Australia.

It is also unclear whether there is any missing data for the chronic diseases in table 1.

I recommend restructuring the discussion such that the strengths and limitations are at the end of the discussion.

The conclusion seems to suggest that there is a difference in contraception use between females with chronic diseases and those without.

The conclusion should address the objectives. What are the patterns of contraceptive use in this cohort? How does chronic disease influence contraceptive use over time?

The recommendations are not necessarily based on the results.

Reviewer #2: Congratulations to all authors! Great work! Background section is very well written/presented in the manuscript. It clearly highlights why this was a much needed research study/analysis. I also like the way the discussion section is framed.

Comments/questions that require authors' attention:

Results section include comparison of groups on various factors and also specifies p-values (bracketed with many findings) to highlight the statistical significance of findings. However, the level of significance (aka. alpha level) at which these p-values were considered to be statistically significant is not clear. For example, was it alpha of 0.05 or 0.1 or 0.3? and why?

Q.1 Are all p-values mentioned under the heading of "Contraceptive use patterns over time", between line 268-275, considered and presented as "Statistically significant" by authors? If yes/no, please clarify (in either case).

Q.2 Line 269-270 states, "women in Status 3 (“sterilisation and other”) in 2006 the most 270 likely to continue using the same contraception in 2012 (P=0.86)". Please clarify how the scientific community/readers of your manuscript should interpret "p=0.86" in terms of statistical significance of a finding stated in line 269-270, to highlight the strength of your evidence/data.

It seems like Table 2 is meant to show results of comparison between two groups of women - "with" and "without" chronic disease in this sample, but it doesn't include p-values or alpha level indicating at which 'level of significance' the found differences were meant to be considered statistically significant. IF available, I would suggest adding the same for all results/tables (where applicable).

Also, consider indicating <0.01 p-values using numbers within the table, rather than using "-" and then adding a note to clarify the meaning of "-", since most of the readers tend to focus on highlighted/important numerical values and ignore '-'.

Thank you, PLOS ONE, for the opportunity to review such a great piece of work by brilliant researchers/authors!

6. PLOS authors have the option to publish the peer review history of their article (what does this mean?). If published, this will include your full peer review and any attached files.

Reviewer #1: No

Reviewer #2: **Yes: **Rushil Acharya

---

## [Author Response · Author response to Decision Letter 0]

3 Apr 2023

We have addressed our response to reviewers in a table which has been uploaded separately.

---

## [Editor Report · Decision Letter 1]

11 Apr 2023

Patterns of contraceptive use through later reproductive years: a cohort study of Australian women with chronic disease

PONE-D-22-13508R1

Dear Dr. Harris,

We’re pleased to inform you that your manuscript has been judged scientifically suitable for publication and will be formally accepted for publication once it meets all outstanding technical requirements.

Kind regards,

Billy Morara Tsima, MD MSc

Academic Editor

PLOS ONE

---

## [Editor Report · Acceptance letter]

24 Apr 2023

PONE-D-22-13508R1 

Patterns of contraceptive use through later reproductive years: a cohort study of Australian women with chronic disease 

Dear Dr. Harris:

I'm pleased to inform you that your manuscript has been deemed suitable for publication in PLOS ONE. Congratulations! Your manuscript is now with our production department. 

Kind regards, 

on behalf of

Dr. Billy Morara Tsima 

Academic Editor

PLOS ONE